# What Is the Role of Traction Test Radiographs in the Preoperative Planning of Adolescent Idiopathic Scoliosis?

**DOI:** 10.3390/jcm12226986

**Published:** 2023-11-08

**Authors:** Giovanni Andrea La Maida, Enrico Gallazzi, Federica Ramella, Marcello Ferraro, Andrea Della Valle, Davide Cecconi, Bernardo Misaggi

**Affiliations:** 1U.O.C. Ortopedia e Traumatologia per le Patologie della Colonna Vertebrale, ASST G. Pini—CTO, 20122 Milano, Italy; giovanniandrea.lamaida@asst-pini-cto.it (G.A.L.M.); marcello.ferraro@asst-pini-cto.it (M.F.); andrea.dellavalle@asst-pini-cto.it (A.D.V.); davide.cecconi@asst-pini-cto.it (D.C.); bernardo.misaggi@asst-pini-cto.it (B.M.); 2Dipartimento di Scienze Biomediche per la Salute, Università degli Studi di Milano, 20122 Milano, Italy; federicaramella@yahoo.it

**Keywords:** AIS, LIV, traction test, side bending, Lenke 1 AIS

## Abstract

Lower instrumented vertebra (LIV) selection is critical to avoid complications like adding-on. This study aims to determine the usefulness of the traction test (TR) in selecting the LIV during surgery for adolescent idiopathic scoliosis (AIS). We analyzed 42 AIS patients with Lenke 1 curves who had preoperative, postoperative, and at least 12-month follow-up X-rays, as well as preoperative side bending (SB) and TR radiograms. Neutral vertebra (NV), stable vertebra (SV), lower instrumented vertebra (LIV), and Cobb angles were identified and compared on all radiographic images. In 23 cases, the TR resulted in SV proximalization compared to the preoperative X-rays, while in 8 cases, SV-TR was more distal. This distalization occurred in 50% of Lenke 1C curves, where a greater correction of the distal curve was found. NV-TR was proximal to NV-preop in 9 cases, while NV-SB was proximal in 22 cases. LIV was proximal to SV-TR in 8 cases, while it was proximal to SV-preop in 22. One patient with LIV proximal to SV-TR developed adding-on. In conclusion, the TR is crucial in AIS preoperative planning as it provides information distinct from that of standard X-rays and SB: (1) it better assesses gravitational stability than rotational stability; and (2) choosing LIV equal to or proximal to SV-TR may prevent adding-on, except in ‘flexible’ Lenke 1C curves where LIV should be equal or distal to SV-preop.

## 1. Introduction

Adolescent idiopathic scoliosis (AIS) surgery aims to correct deformity and achieve the most balanced spine that a patient can tolerate in the coronal, axial, and sagittal planes. Additionally, the goal of the surgeon is to achieve a balanced spine that remains stable over time, preventing long-term complications, such as adding-on (AO), proximal junctional kyphosis (PJK), and distal junctional kyphosis (DJK). In particular, AO has been demonstrated to be influenced by several factors, such as: (1) younger age and less skeletal maturity; (2) smaller preoperative proximal thoracic, main thoracic, and lumbar curves; (3) larger preoperative coronal and sagittal imbalances; (4) an overcorrection of coronal and sagittal alignments; and (5) selection of the lowest instrumented vertebra (LIV) [1]. Indeed, the correct selection of the LIV appears to be critical in preventing AO [2]. Several methods for LIV selection have been described in the literature, with most of them being based on the evaluation of the preoperative gravitational and rotational stability of the hypothetical LIV via the evaluation of full-spine standing X-rays and side bending [3]. Nonetheless, the prevalence of AO remains high, with reported rates of up to 20% in some series [4,5].

The role of the Risser table traction test (TR) has been widely debated over the last 25 years. Traditionally, the use of the TR has been limited to cases in which the execution of voluntary side bending was difficult, such as in young children or patients with neuromuscular etiologies. On the contrary, its usefulness in the diagnostic workup and preoperative planning of AIS has largely been questioned. While the reliance on side bending to evaluate curve flexibility is common among most centers, several papers, even recently, investigated the potential usefulness of the TR in surgical planning [6,7]. This is especially true for Lenke 1 curves, where the TR is considered useful by some authors for stiffer, larger curves with a Cobb angle > 70° [8]. Indeed, the usefulness of the TR is based largely on expert opinion, with some recent papers questioning it by focusing on the comparison between the TR and side bending in evaluating curve flexibility [6]; however, no study has evaluated the overall changes in the rotational and gravitational stability of Lenke 1 curves by describing the variation in neutral and stable vertebrae with the TR. Furthermore, most of the previous studies did not report a standardized method for performing the TR.

Therefore, it is still unclear whether the TR offers additional rather than competitive information with side bending (SB), and whether it allows for a better assessment of the rotational or gravitational stability of the curve. In this context, our hypothesis is that the TR could provide additional rather than competitive information on curve behavior; therefore, the purpose of this study is to evaluate the behavior of Lenke 1 curves with the TR, regarding the variation in NV and SV, to identify useful parameters for the choice of LIV.

## 2. Materials and Methods

After receiving institutional review board (IRB) approval for the retrospective evaluation of the medical data and radiographic information of patients with AIS, a search on the institutional database was performed to identify patients with the following inclusion criteria: a diagnosis of Lenke 1-type AIS treated surgically between 2015 and 2021; availability of full standing spine X-rays preoperatively, postoperatively, and at a 12-month minimum follow-up; and availability of preoperative bending and TR radiograms. Patients diagnosed with a secondary form of scoliosis, with an age <10 years or >18 years at the index surgery, or with incomplete FU data were excluded from the study.

TR and SB are routinely performed in a standardized manner in our institution at a maximum of 3 days before surgery. SB is performed supine by an experienced technician with a specific formation, asking for the maximum bending achievable by the patient on each side on a single X-ray for each side. The TR was performed in a standard manner for all included patients, with the chin traction on Risser’s frame corresponding to 50% of their body weight, with a maximum of 30 kg. The test was performed in the ward with standard equipment and without the requirement for anesthesia (Figure 1). Three experienced orthopedic surgeons evaluated both the preoperative full-spine X-rays, SB and TR, and independently measured the Cobb angles for the main and secondary curves. Furthermore, the NV and SV were identified on the preop X-rays, SB and TR, and any disagreement between each evaluator was solved with a collegial discussion.

Each patient underwent surgery with the correction and instrumentation technique used in our institution, which consists of an apical derotation and posterior translation with a hybrid instrumentation with screws and sublaminar bands.

LIV was evaluated on postoperative X-rays, as well as the magnitude of curve correction. The presence of adding-on was defined as >3 cm of radiographic progression of the LIV to the CSVL distance or an >10 increase in the coronal disc angle below the LIV at the final FU [9,10].

The statistical analyses were performed with a freely available statistical calculation tool (VassarStats: Statistical Calculation Website. Available at www.vassarstats.net (accessed on 22 July 2023). The continuous variables were expressed as means ± standard deviations. Student’s *t*-test for paired samples was used to compare pre- and postoperative radiological parameters. Chi-square was used to compare the percentages. A value of *p* ≥ 0.05 was considered significant in all tests.

## 3. Results

Of the 210 patients retrieved from the database search, 42 (9 M, 33 F, mean age 15.1 years) met the inclusion criteria and thus were included in the study. Of these, 20 were Lenke 1A, 6 were Lenke 1B, and 16 were Lenke 1C. The mean Cobb angle of the main curve was 56.7° ± 10.9° in preop and 17.8° ± 10.8° in postop (*p* < 0.01, *t*-test).

The TR was well tolerated by all included patients; no interruption of the test was required, and the neurological examination after the test was normal for all the patients.

A comparison between the TR and preop X-rays was conducted concerning the gravitational stability of the curve, while the rotational stability was evaluated on the TR, SB, and preop X-rays. The gravitational stability was assessed by evaluating the SV, while the rotational stability was assessed by evaluating the NV. Regarding the gravitational stability, in 23 cases (54.8%), we observed a proximalization of the SV at the TR with respect to the preop X-rays (Figure 2; mean + 2.4 ± 1.2 levels); in 11 cases, we observed no modification of the SV; while in 8 cases, the SV-TE was more distal than the SV-preop. The ‘distalization’ of the SV-TR was observed only in the Lenke 1C curves, (8 out of 16 cases). In those cases with distalization of SV, a significantly greater correction of the Cobb angle of the distal curve was observed both with SB and TR (SB: 62.2 ± 8.8% vs. 49.5 ± 12.9%, *p* = 0.05, *t*-test; TR 60.1 ± 6.5% vs. 44.6 ± 10.7%, *p* = 0.05, *t*-test) (Figure 3).

Concerning the rotational stability, the NV-TR was proximal to the NV-preop in only 9 cases (21.4%, mean + 1.2 ± 0.7 levels; *p* = 0.06, chi-square test for the proximalization of the NV-TR vs. SV-TR); in contrast, the NV was proximal in 22 (52.4%) of the cases in the SB (*p* < 0.05, chi-square test). When analyzing the LIV on the postop X-rays, we observed that in 20 cases (47.6%), the LIV was proximal to the SV-preop, while it was proximal to the SV-TE in only 8 (19%) (*p* < 0.05, chi-square test). One patient with the LIV proximal to the SV-TE developed radiographic adding-on, while none of the patients with the LIV distal to the SV-TE developed adding-on.

## 4. Discussion

Side bending and the TR are two different preoperative radiographic tests used in the surgical planning for AIS. While the two aim to evaluate similar things, namely curve flexibility and rotational and gravitational stability, it is still unclear whether the TR offers additional rather than competitive information with SB. Thus, our working hypothesis was that the TR could provide additional rather than competitive information on curve behavior. In this context, the main findings of this paper were that the TR is crucial in AIS preoperative planning as it provides information distinct from that of standard X-rays and SB: (1) it better assesses gravitational stability than rotational stability; and (2) choosing LIV equal to or proximal to SV-TR may prevent adding-on, except in ‘flexible’ Lenke 1C curves where LIV should be equal or distal to SV-preop

Multiple preoperative radiographic methods have been used to establish the correct extension of AIS surgery instrumentation in order to balance the need for motion-segment preservation while maximizing the reduction in the risk of long-term complications, such as DJK and adding-on. Historically, the methods for LIV selection varied according to the surgical technique used for curve correction, with a progressive reduction in the fusion area. Indeed, while in the Risser era, the fusion extended from end-to-end vertebrae with neutral rotation, and with the introduction of the Cotrel–Dubousset instrumentation, a more proximal LIV was usually selected. Moreover, the most recent literature has highlighted the importance of evaluating both the gravitational and rotational stability of the main curve, as well as the flexibility of the compensatory curves [3,11,12].

In main thoracic curves (Lenke 1), the gravitational and rotational stability of the curves are assessed by identifying the neutral vertebra (NV) and the stable vertebra (SV) in the preoperative standing full-spine X-rays and evaluating the reducibility of the lumbar curve on the standing side-bending radiograph. In this setting, the role of traction test radiographs remains poorly defined. In a recent survey of Japanese orthopedic surgeons, it was found that SB is usually preferred over the TR for evaluating curves with a Cobb angle between 40° and 70°, whereas TR radiography is preferred over SB for rigid and larger curves, usually with a Cobb angle between 70° and 90° [8]. This preference was derived from early studies that compared SB and the TR with regard to their ability to evaluate curve flexibility. Overall, the TR was considered superior to SB in evaluating flexibility when curves were more severe, shorter, and stiffer [13,14]. Following these observations, and with the aim of better defining the role of the TR in AIS preoperative evaluations, several papers have compared the TR with SB regarding the ability to better define curve flexibility. Watanabe et al. compared the two radiographic techniques and demonstrated that the TR could be superior to side bending in evaluating the curve flexibility in main thoracic curves under some strict conditions. These conditions were a Cobb angle > 60°, an age < 15 years, an apex located between T4 andT8/T9, a normal kyphosis, and no more than seven involved vertebrae. In contrast, they found that the majority of thoracolumbar/lumbar curves were corrected more effectively using side-bending radiographs [15]. Davis first described traction under general anesthesia (TRUGA), which allows the application of maximal axial distraction and translation. He found TRUGA to be superior to supine-bending radiography in assessing curve flexibility before surgery [16]. Similarly, Tokala et al. compared fulcrum bending with TRUGA. They found that TRUGA was more predictive of curve correction than fulcrum bending in cases where AIS was corrected with pedicle screw constructs [17]. However, this approach has an important limitation: while anesthesia allows the complete avoidance of patient contraction and spasms, it is less convenient for surgical planning since the test must be performed just before surgery, thus preventing true preoperative planning.

Nonetheless, the utility of the TR in preoperative planning was still under question. In this context, two recent studies compared the TR with SB with regard to their usefulness in LIV selection. Arima et al. evaluated the theoretical preoperative planning of 30 AIS cases via review by 32 spine surgeons, who reviewed each case three times: the first time with preop X-rays and SB, the second time with preop X-rays and the TR, and the third time with all the methods available. Overall, they found that whether raters used SB, TR, or both in addition to preop X-rays, this did not influence the decision-making for LIV in AIS Lenke type 1 surgery [6]. Malik et al. investigated whether the TR and SB yielded the same Lenke classification, given that the judgment of curve structurality is a major factor in defining fusion levels. They found that the TR and SB provided a concordant Lenke classification 82.1% of the time. Interestingly, in discordant cases, the TR underestimated the TL/L curve correction, resulting in a higher (and potentially wrong) Lenke classification. Therefore, they concluded that the TR was not an adequate substitute for SB. Again, these data seem to indicate that the TR is not useful, or at best, does not have a clearly defined role in AIS preoperative planning [18].

It is, however, important to highlight that none of the previously cited studies detailed the kind of information that could be extracted from the TR. Recently, Fischer et al. introduced the concepts of gravitational stability (GS), defined as the relationship between the central sacral vertical line (CSVL) and the SV, and rotational stability (RS), defined as the NV. They demonstrated that curve progression after surgery increases significantly when both the GS and RS are not respected [3]. It is yet to be clarified whether the TR allows for a better evaluation of the RS and the GS of the curve. According to our data, the TR has a significantly greater effect on the SV than on the NV, thus providing more information on the gravitational stability of the curve. Indeed, we selected an LIV proximal to the SV-TR in only a minority of cases (8%) and, interestingly, the only failure we observed at 2 years’ FU was in the case where the LIV was proximal to the SV-TR, thus hypothetically highlighting that the fusion area was too short. This observation was previously reported in early-onset scoliosis, with the concept of ‘stable-to-be’ vertebra [19]: according to this concept, the vertebra most closely bisected by the CSVL on the TRUGA or concave bending was the ‘stable-to-be’; when the LIV was at or distal to the ‘stable-to-be’ vertebra, only minimal distal disk wedging (i.e., adding-on) was observed, similarly to what we report in this paper. Interestingly, a recent paper from Kim et al. reported a similar observation without the use of traction: the authors investigated the GS on supine X-rays without traction and identified cases in which the last substantially touched vertebra (LSTV) was proximalized in supine X-rays when compared to standing X-rays. In those cases where the supine LSTV was chosen as the LIV, the outcomes were the same as the cases where the LIV was the standing LSTV [20]. Both those approaches—with or without traction—evaluate curve flexibility in different conditions and functionally allow for fusing more proximally to the standing LSTV, sparing motion segments. Another recent paper highlighted that the risk factors for failure when fusing at LSTV-1 are skeletal immaturity, coronal imbalance, and excessive deviation of the LSTV from the CSVL, while curve flexibility was not a risk factor [21]. On the contrary, we observed little effect of the TR on axial rotation, with the NV-TR equal to the NV-preop in 79% of cases. This observation seems to contrast with previous studies, where the TR was superior to SB in reducing curve rotation. However, in these studies, rotation was measured on the apical vertebra; no study has evaluated NV shift, which is more useful for preoperative planning [7,22]. Another important observation that we reported is the paradoxical effect of the TR on the SV in the subset of Lenke 1C curves with a flexible TL compensatory curve. In these cases, we observed the distalization of the SV-TR with respect to the SV-preop, which was due to the high flexibility of the compensatory curve, as shown by the greater Cobb angle correction with SB, which tends to straighten with traction. These curves might be less gravitationally stable due to the flexible compensatory curve, and thus it is the authors’ opinion that they should be fused to or distal to the SV-preop. Finally, a strength of this study is that we report a standardized and reproducible method for TR execution. The only study that proposes a standardization for the TR is a recent paper by O’Neill et al. [7]; the authors experimentally tested the use of a traction force of 1.5 of the patient’s body weight, while in our paper, we applied a force that is only 0.5 of the patient’s body weight, to a maximum of 30 kg. To apply such a larger force, in the other study, they used a specifically manufactured traction table, to allow for the minimum duration of traction (5 s). In our study, traction was applied with the use of a standard and widely available Risser frame and kept for longer to allow the technician to obtain the X-rays. With this technique, we did not observe any complications or discomfort for the patients. It is still unclear whether a higher traction force could provide more information to the surgeon than a lower traction force.

This study has several limitations. First, the small sample size may have prevented the observation of more long-term complications, thus hampering the interpretation of our data in the context of adding-on prevention. Nonetheless, we believe that our findings are consistent enough to provide relevant information regarding the LIV choice in Lenke 1 curves. Secondly, no other study evaluated the shift of the SV and NV with different radiographic methods reported here; thus, our findings need to be externally confirmed by other studies. Finally, since there is no standardization for TR methods of execution across the literature, it could be difficult to compare our findings to previous reports, as discussed above.

## 5. Conclusions

In conclusion, the main results of this study can be summarized as follows:The TR allows for better assessment of the gravitational stability of scoliotic curves than the rotational stability, as demonstrated by the greater effect on SV than NV;The choice of an LIV equal to or distal to SV-TR could be protective against adding on, even in cases where this is proximal to ST-preop;In ‘flexible’ Lenke 1C curves, the TR may have a paradoxical effect with the distalization of the SV, due to overcorrection of the compensation curve; in these cases, the choice of LIV should be equal or distal to the SV-preop.

Therefore, the TR definitely has a role in the preoperative planning of AIS, providing different information than preop and bending X-rays.

## Figures and Tables

**Figure 1 jcm-12-06986-f001:**
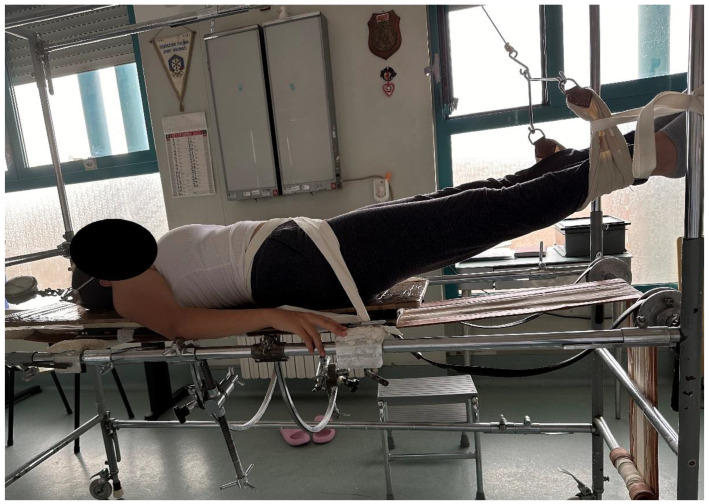
Picture showing how the patient is positioned for examination in our institution. Chin traction is applied up to 30 kg or 50% of the body weight. The X-ray tube (not shown) is inserted from the side of the bed.

**Figure 2 jcm-12-06986-f002:**
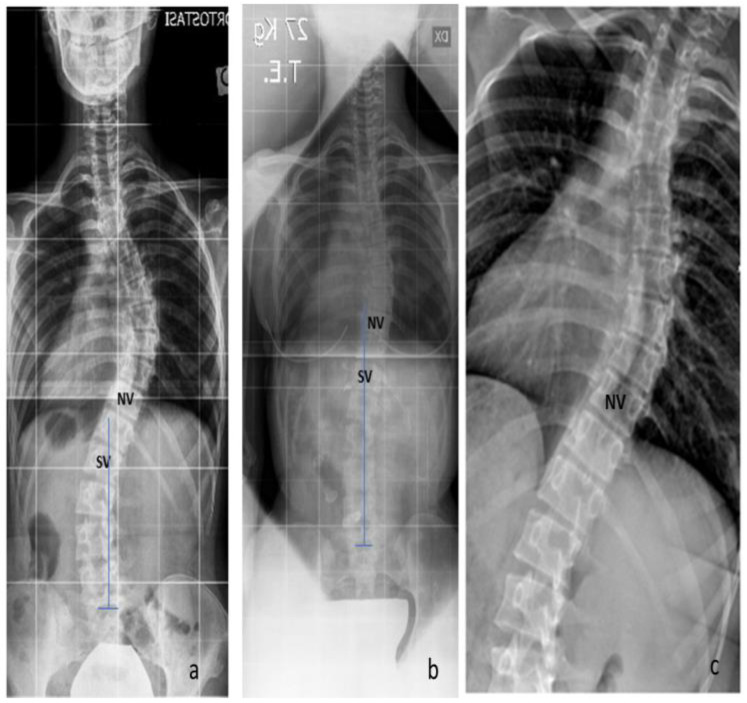
Gravitational stability: the proximalization of the SV at the TR with respect to the preop X-rays. (**a**) The SV-preop is L1; (**b**) the SV-TR is T12; (**c**) the SB shows the proximalization of the NV.

**Figure 3 jcm-12-06986-f003:**
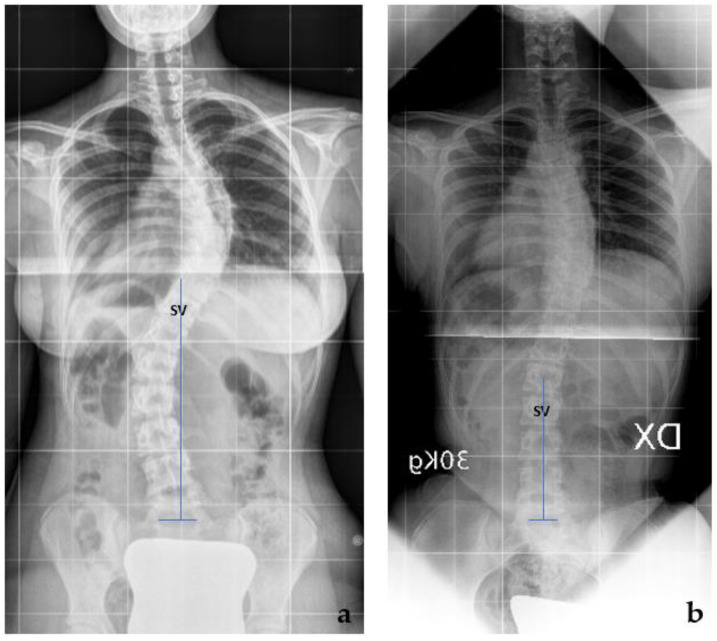
Lenke 1C curves: in 8 cases, the SV-TE was more distal than the SV-preop. (**a**) The SV-preop is T11; (**b**) the SV-TR is L2.

## Data Availability

Data for this study are available on request to the corresponding author.

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
