# Peer review of "What Is the Role of Traction Test Radiographs in the Preoperative Planning of Adolescent Idiopathic Scoliosis?"

_jcm, 2023, doi:10.3390/jcm12226986_

Round 1
Reviewer 1 Report
Comments and Suggestions for Authors
- The authors performed an original study on the role of traction test radiographs in preoperative planning of adolescent idiopathic scoliosis. The study results have shown that TR is crucial in AIS preoperative planning by providing distinct information from standard X-rays and SB: (1) it better assesses gravitational stability than rotational stability (2) choosing LIV equal to or proximal to SV-TR may prevent adding-on, except (3) in ‘flexible’ Lenke 1C curves where LIV should be equal or distal to SV-Preop.
Title
- Please write “adolescent idiopathic scoliosis” before using its abbreviated form.
Abstract
- There are some abbreviations that are not defined while using them for the first time (e.g., NV, SV, SV-TE,
- Introduction
o Lines 44-47 needs supporting references.
- Methods
o Why the authors did not obtain supine lateral bending radiograph that is routinely considered for pre-operative evaluation of the scoliosis curves?
o Please provide an example radiograph of lateral bending test.
o There is no information regarding statistical analyses.
- Results
o The authors stated that “Regarding the gravitational stability, in 23 cases (54.8%) we observed a proximalization of the SV at the TR in respect to the Preop X-rays”. Please provide the values of proximalization.
o Again, “Concerning rotational stability, the NV-TR was proximal to the NV-Preop in only 9 cases (21.4%),” what was the distance between NV-TR and NV-pre-operation?
o There is no information about PJK.
- Discussion:
o In the first paragraph of discussion section, the authors should provide the main findings of the study.
Comments on the Quality of English Language
- There are some grammatical and syntax errors in the text.
Author Response
Dear Reviewer 1,
First of all, thank you very much for your comments and for the opportunity to improve our paper.
The authors performed an original study on the role of traction test radiographs in preoperative planning of adolescent idiopathic scoliosis. The study results have shown that TR is crucial in AIS preoperative planning by providing distinct information from standard X-rays and SB: (1) it better assesses gravitational stability than rotational stability (2) choosing LIV equal to or proximal to SV-TR may prevent adding-on, except (3) in ‘flexible’ Lenke 1C curves where LIV should be equal or distal to SV-Preop.
Title
- Please write “adolescent idiopathic scoliosis” before using its abbreviated form.
Thank you for your remark, corrected in text
Abstract
- There are some abbreviations that are not defined while using them for the first time (e.g., NV, SV, SV-TE,
Thank you for your remark. Corrected in text
Introduction
- Lines 44-47 needs supporting references.
Thank you for your remark. We added the correct references.
Methods
- Why the authors did not obtain supine lateral bending radiograph that is routinely considered for pre-operative evaluation of the scoliosis curves?
Thank you for your question. We do obtain supine SB for each patient pre-operatively, as stated in the methods (Line 74-77). We corrected a mistake in the text.
- Please provide an example radiograph of lateral bending test.
We modified Figure 1 to show a SB X-Rays that shows the proximalization of the NV in the case shown
- There is no information regarding statistical analyses.
Thank you for your remark, we added a ‘Statistical Analysis’ section in the Methods.
Results
- The authors stated that “Regarding the gravitational stability, in 23 cases (54.8%) we observed a proximalization of the SV at the TR in respect to the Preop X-rays”. Please provide the values of proximalization.
Thank you for your remark. We added in the text the mean ± SD level of proximalization of the SV at the TR.
- Again, “Concerning rotational stability, the NV-TR was proximal to the NV-Preop in only 9 cases (21.4%),” what was the distance between NV-TR and NV-pre-operation?
Thank you for your remark. We added in the text the mean ± SD level of proximalization of the NV at the TR.
- There is no information about PJK.
Thank you for your question. PJK evaluation was beyond the scope of this paper. Nonetheless, we reviewed the cases included in the study, and no clinical or radiological PJK were observed at the lastest Follow-up.
Discussion:
- In the first paragraph of discussion section, the authors should provide the main findings of the study.
Thank you for your comment. We summarized the main findings at the beginning of the discussion as requested.
Reviewer 2 Report
Comments and Suggestions for Authors
Dear authors,
thank you for your contribution.
There are few weaknesses:
-it is low case volume
-there is a lot of literature regarding how to choose LIV in all Lenke scolioses types published after 2020, which is not included in your paper. Many citations are older... please update and discuss accordingly.
-what do you think of "stable to be" vertebra concept?
-it is true that not many use TR but rather SD: so please go more into detail in methods how precisley you do it at your institution. Saying this because Risser frames are hard to find these days, at least in my country. Is it performed horizontally, therefore without gravitation effect? If so, what does it mean? Maybe a photo?
-explain abbreviation TE
Author Response
Dear Reviewer 2,
First of all, thank you very much for your comments and for the opportunity to improve our paper.
Dear authors, thank you for your contribution.
There are few weaknesses:
- it is low case volume
Thank you for your remark. We are aware of the limited sample of this study, as we highlighted in the limitation of the study (See Discussion, Line 211-213)
-there is a lot of literature regarding how to choose LIV in all Lenke scolioses types published after 2020, which is not included in your paper. Many citations are older... please update and discuss accordingly.
Thank you for your comment. We reviewed the most recent literature and added relevant papers to our discussion.
-what do you think of "stable to be" vertebra concept?
Thank you very much for this interesting question. We re-evaluated literature on the ‘stable to be’ vertebra, and we found that this concept is very similar to what we describe in our paper. Our feeling is that either the method that we describe in our paper and the ‘stable to be’ concept are aimed at safely proximalizing the LIV, thus sparing motion segments without increasing the risk for adding on; interestingly, in a paper from Dede et al (J Ped Orthop, 2016, doi 10.1097/BPO.0000000000000467) when LIV was at or distal to the ‘stable to be’ only minimal distal disk wedging (i.e. adding on) was observed, similarly to what we report in this paper. We added this to the Discussion
-it is true that not many use TR but rather SD: so please go more into detail in methods how precisley you do it at your institution. Saying this because Risser frames are hard to find these days, at least in my country. Is it performed horizontally, therefore without gravitation effect? If so, what does it mean? Maybe a photo?
Thank you for your question. We added a picture that shows how the TR is performed in our Institution. We hope that now the description is clearer.
-explain abbreviation TE
Thank you for your remark, it was a spelling mistake now corrected in text.
Reviewer 3 Report
Comments and Suggestions for Authors
The authors determine the usefulness of a traction test (TR) in choosing a LIV
during surgery for adolescent idiopathic scoliosis (AIS).
The purpose of the study is very interesting and the clinical significance is very large. However, the number of cases is small, and analysis and amount of data are insufficient.
Author Response
Dear Reviewer 3,
Thank you very much for your time and your comment.
The authors determine the usefulness of a traction test (TR) in choosing a LIV
during surgery for adolescent idiopathic scoliosis (AIS).
The purpose of the study is very interesting and the clinical significance is very large. However, the number of cases is small, and analysis and amount of data are insufficient.
Thank you for your remark. We are aware of the limited number of patients included in our study, as we stated in the Discussion, Line 211-213. Nonetheless, the number of cases included in our paper is on par with most of the similar paper published on this subject. Furthermore, despite aware of this limitation, we think that our results are nonetheless interesting for our community. We would be grateful if you could consider again our paper for publication.
Round 2
Reviewer 1 Report
Comments and Suggestions for Authors
The authors addressed the most of my previous comments. However, there are some others that did not address:
1. ​ The authors did not pay attention to my comments for title and abstract. If they do not agree with these comments, please leave a reasonable reason to reject them.
2. Line 73: What is the meaning of SB?
3. My previous comment regarding the position of imaging was:
- "Why the authors did not obtain supine lateral bending radiograph that is routinely considered for pre-operative evaluation of the scoliosis curves?" and the authors' response was:
- "We do obtain supine SB for each patient pre-operatively, as stated in the methods (Line 74-77)."
However, in line 73 it has been written SB are performed standing. Please clarify this discrepancy.
4. The lateral bending position of the patient in figure 1 is not visible.
4. Line 94: Statistical value of less than 0.05 should be considered as statistical difference.
Comments on the Quality of English LanguageThe quality of English is satisfactory.
Author Response
Dear Reviewer,
Thank you very much for your time and effort in reviewing our paper.
The authors addressed the most of my previous comments. However, there are some others that did not address:
- ​ The authors did not pay attention to my comments for title and abstract. If they do not agree with these comments, please leave a reasonable reason to reject them.We are deeply sorry, this was a mistake on our side. We changed the title and abstract accordingly.
2. Line 73: What is the meaning of SB?
SB is the abbreviation for Side Bending. We extended this in line 60. Again, sorry for the confusion
3. My previous comment regarding the position of imaging was:
- "Why the authors did not obtain supine lateral bending radiograph that is routinely considered for pre-operative evaluation of the scoliosis curves?" and the authors' response was:
- "We do obtain supine SB for each patient pre-operatively, as stated in the methods (Line 74-77)."
However, in line 73 it has been written SB are performed standing. Please clarify this discrepancy.
Sorry, again a mistake from our side. The SB are performed supine as in our previous comment. We corrected the text.
4. The lateral bending position of the patient in figure 1 is not visible.
The SB is in Figure 1. Due to a space constrain we only put in the Figure one side, the one useful for the Analysis.
4. Line 94: Statistical value of less than 0.05 should be considered as statistical difference.
Again, writing mistake from our side. We corrected the text.
